# Treatment of necrotizing enterocolitis by conditioned medium derived from human amniotic fluid stem cells

**Joshua S. O'Connell**[1,2‡], **Bo Li**[1,2‡], **Andrea Zito**[1,2], **Abdalla Ahmed**[1,2],
**Marissa Cadete**[1,2], **Niloofar Ganji**[1,2], **Ethan Lau**[1,2], **Mashriq Alganabi**[1,2], **Nassim Farhat**[1,2],
**Carol Lee**[1,2], **Simon Eaton**[3], **Robert Mitchell**[4], **Steve Ray**[4], **Paolo De Coppi**[3], **Ketan Patel**[4],
**Agostino Pierro**[1,2]*

**1** Division of General and Thoracic Surgery, The Hospital for Sick Children, Toronto, Ontario, Canada,
**2** Translational Medicine Program, The Hospital for Sick Children, Toronto, Ontario, Canada, **3** Stem Cells &
Regenerative Medicine Section, NIHR BRC Great Ormond Street Hospital and UCL Great Ormond Street
Institute of Child Health, London, United Kingdom, **4** Micregen Ltd, Thames Valley Science Park, Reading,
United Kingdom

‡ JSO and BL authors contributed equally to this work and share first authorship.
* agostino.pierro@sickkids.ca

Biotechnology, ITALY

**Data Availability Statement:** All relevant data are
within the paper and its Supporting Information
files.

## Abstract

### Purpose

Necrotizing enterocolitis (NEC) is one of the most distressing gastrointestinal emergencies
affecting neonates. Amniotic fluid stem cells (AFSC) improve intestinal injury and survival in
experimental NEC but are difficult to administer. In this study, we evaluated whether condi-
tioned medium (CM) derived from *human* AFSC have protective effects.

### Methods

Three groups of C57BL/6 mice were studied: (i) breast-fed mice as control; (ii) experimental
NEC mice receiving PBS; and (iii) experimental NEC mice receiving CM. NEC was induced
between post-natal days P5 through P9 via: (A) gavage feeding of hyperosmolar formula
four-time a day; (B) 10 minutes hypoxia prior to feeds; and (C) lipopolysaccharide adminis-
tration on P6 and P7. Intra-peritoneal injections of either PBS or CM were given on P6 and
P7. All mice were sacrificed on P9 and terminal ileum were harvested for analyses.

### Results

CM treatment increased survival and reduced intestinal damage, decreased mucosal
inflammation (IL-6; TNF-α), neutrophil infiltration (MPO), and apoptosis (CC3), and also
restored angiogenesis (VEGF) in the ileum. Additionally, CM treated mice had increased
levels of epithelial proliferation (Ki67) and stem cell activity (Olfm4; Lgr5) compared to NEC
+PBS mice, showing restored intestinal regeneration and recovery during NEC induction.
CM proteomic analysis of CM content identified peptides that regulated immune and stem
cell activity.

**Funding:** AP is the recipient of a Canadian Institutes of Health Research (CIHR) Foundation Grant #353857 and the Robert M. Filler Chair of Surgery, The Hospital for Sick Children. The funding institutes had no effect on study design, analysis, or interpretation of results.

**Competing interests:** RM, SR, PD, and KP are stockholder of Microgen Ltd, this relationship had no effect on study design, analysis, or interpretation of results, and this does not alter our adherence to PLOS ONE policies on sharing data and materials. All other authors declared they had no conflict of interest.

## Conclusions

CM derived from human AFSC administered in experimental NEC exhibited various benefits including reduced intestinal injury and inflammation, increased enterocyte proliferation, and restored intestinal stem cell activity. This study provides the scientific basis for the use of CM derived from AFSC in neonates with NEC.

## Introduction

Necrotizing enterocolitis (NEC) is one of the most devastating gastrointestinal diseases affecting preterm, low birth weight, and very low birth weight neonates [1]. NEC is characterized by intestinal mucosal injury, inflammation, and necrosis often leading to perforation [1]. Although modern medicine and critical care have advanced, treatment for NEC remains unchanged. The mortality from NEC has remained unchanged for decades and reported to be as high as 30–50% [2]. Furthermore, morbidity for NEC survivors remains high including intestinal failure and neurodevelopmental delay [3]. Current medical treatment of suspected or confirmed NEC is supportive unless further deterioration is seen. If this is the case, an operation remains the only option [1]. It is generally agreed that the best treatment practice would involve the development of a medical treatment to avoid an operation and intestinal resection [1].

Regenerative medicine has the potential to improve treatment for NEC. The two most common stem cell lineages used in NEC have been amniotic fluid stem cells (AFSC) [4–7] and postnatal bone marrow-derived mesenchymal stem cells (BM-MSC) [6–8]. These cells have been studied in animal models to assess their potential effects against the development of experimental NEC [5, 9]. Previous studies have shown that both AFSC and BM-MSC reduce intestinal damage and inflammation in rodent pups exposed to experimental NEC [6, 7]. It has been reported that AFSC is superior to BM-MSC due to being easier to obtain and expand during cellular culture [4, 6, 10]. AFSC administered during experimental NEC integrate at a low frequency into the bowel wall, but nevertheless improved survival, reduced the incidence of NEC, decreased gut damage, and improved intestinal absorption [4]. Although stem cell treatment has potential, its clinical use may be difficult because of potential tumorgenicity, immunogenicity, and other detrimental effects to the host [9]. Recent studies in other diseases have indicated the potential benefit of the administration of conditioned medium (CM) cultured from stem cells as an alternative to AFSC administration [11–13]. However, it is unknown whether NEC could be mitigated by the administration of CM cultured from AFSC.

We hypothesized that the administration of human-derived AFSC conditioned medium (hAFSC-CM) would lower NEC associated damage compared to those treated with a placebo (phosphate buffered saline (PBS) injected). As the terminal ileum is the most affected area of NEC, we studied this portion of the intestine to evaluate pup's survival, intestinal morphology, injury, intestinal proliferation, and stem cell activity in response to hAFSC-CM.

## Materials and methods

### Human-derived amniotic fluid stem cell conditioned medium

Conditioned medium derived from human amniotic fluid stem cells (hAFSC-CM) was provided at no cost from Microgen Ltd. (London, United Kingdom). The complete process and characterization of the conditioned medium is described in the manuscript by Mellows et al.

[13]. Briefly, hAFSCs were pelleted in 1.5ml microfuge tubes by centrifugation at 300g at a density of $1x10^6$ cells/tube and incubated in 400µl sterile PBS/tube for 24 hours. Following incubation, the supernatant was aspirated, pooled and centrifuged at 2,000g for 20 minutes.

CM was processed according to the Filter Aided Sample Preparation protocol (FASP). All steps were performed on 10kD MWCO filters (Vivacon 500). Proteins were denatured in 8M urea, 100mM ammonium bicarbonate, pH 8 (buffer D) and reduced in the presence of 1mM dithiothreitol for 30 min at room temperature. The buffer was changed to buffer D containing 5.5mM iodoacetamide and shaken at room temperature for 30 min in the dark to alkylate proteins. Samples were then washed twice with 100µl buffer D and twice with 100µl 100 mM ammonium bicarbonate, pH 8 (ABC buffer). To digest proteins, samples were incubated overnight with 2.4µg trypsin. Next day, peptides were collected by centrifugation and filters were washed with 80µl ABC buffer. Both eluates were combined and acidified with trifluoroacetic acid to a final concentration of 1%. Peptides were purified/desalted using C18-stage tips.

LC-MS/MS measurements were performed on a QExactive Plus mass spectrometer (Thermo Scientific) coupled to an EasyLC1000 nanoflow-HPLC. HPLC–column tips (fused silica) with 75µm inner diameter were self-packed with Reprosil-Pur 120 C18-AQ, 1.9µm (Dr. Maisch GmbH) to a length of 20 cm. Samples were applied directly onto the column without a pre-column. A gradient of A (0.1% formic acid in water) and B (0.1% formic acid in 80% acetonitrile in water) with increasing organic proportion was used for peptide separation (loading of sample with 0% B; separation ramp: from 5–30% B within 85 min). The flow rate was 250nL/min and for sample application 600nL/min. The mass spectrometer was operated in the data-dependent mode and switched automatically between MS (max. of $1x10^6$ ions) and MS/MS. Each MS scan was followed by a maximum of ten MS/MS scans using normalized collision energy of 25% and a target value of 1000. Parent ions with a charge state form z = 1 and unassigned charge states were excluded for fragmentation. The mass range for MS was m/z = 370–1750. The resolution for MS was set to 70,000 and for MS/MS to 17,500. MS parameters were as follows: spray voltage 2.3kV; no sheath and auxiliary gas flow; ion-transfer tube temperature 250˚C.

The MS raw data files were uploaded into the MaxQuant software version 1.6.2.10 for peak detection, generation of peak lists of mass error corrected peptides, and for database searches. A full-length UniProt human database additionally containing common contaminants, such as keratins and enzymes used for digestion, was used as reference. Carbamidomethyl cysteine was set as fixed modification and protein amino-terminal acetylation and oxidation of methionine were set as variable modifications. Three missed cleavages were allowed, enzyme specificity was trypsin/P, and the MS/MS tolerance was set to 20 ppm. The average mass precision of identified peptides was in general less than 1 ppm after recalibration. Peptide lists were further used by MaxQuant to identify and relatively quantify proteins using the following parameters: peptide and protein false discovery rates, based on a forward-reverse database, were set to 0.01, minimum peptide length was set to 7, minimum number of peptides for identification and quantitation of proteins was set to one which must be unique. The 'match-between run' option (0.7 min) was used.

The cellular component gene ontology and functional enrichment analysis was performed using g:Profiler (version e104_eg51_p15_3922dba) [14]. Cytoscape was used for network visualization with the Enrichment Map and AutoAnnotate plugins as previously described [15].

## Experimental mouse model

Following ethical approval by the Animal Care Committee at The Hospital for Sick Children (protocol# 44032), we used a previously described mouse model of NEC [16]. C57BL/6 mouse

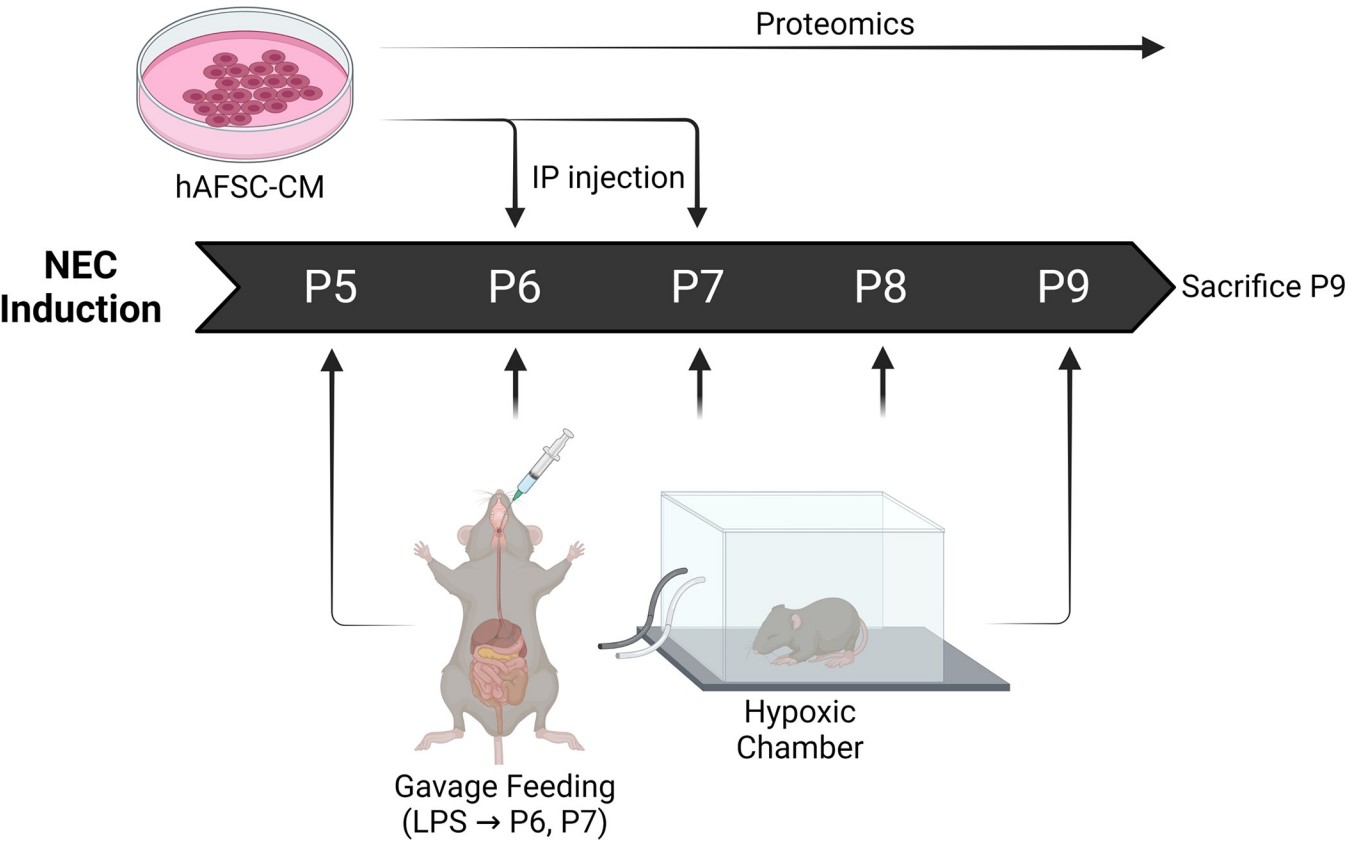

**Fig 1. Schematic illustration of experimental NEC model and CM administration.** From postnatal days (p) 5 to p9, mouse pups exposed to NEC induction, which including four times daily hyperosmolar formula feeding and hypoxia, and LPS administration on p6 and p7. AFSC-CM were collected, assessed by proteomics, and administered by intraperitoneal injection to the mice pups undergoing NEC induction on p6 and p7.

pups who were post-natal day 5 (P5) were divided into three groups (**Fig 1**): (i) control mice who were breast fed; (ii) mice exposed to experimental NEC and phosphate buffered saline (PBS) administration (NEC+PBS); and (iii) mice exposed to experimental NEC and hAFSC-CM administration (NEC+CM). NEC was induced starting on P5 through P9 via: (A) hyperosmolar formula given through a gavage feeding tube (15g Similac mixed with 75mL Esbilac) four times daily; (B) 10 minutes within a hypoxic chamber (5% $O_2$, 5% $CO_2$, 90% $N_2$) prior to all feeds; and (C) lipopolysaccharide (LPS; 4mg/kg/day) administration through a gavage feeding tube on the mornings of P6 and P7, as previously described [17, 18]. Injections were given intra-peritoneally (100 μL) of either PBS (control) or CM (treatment) on P6 and P7. CM dose and administration period was based on previous literature showing therapeutic effects of AFSC's in NEC and hAFSC-CM [12, 13, 19, 20]. On P9, all mice were sacrificed by cervical decapitation and terminal ileum was harvested. Ileal histology, inflammation, intestinal proliferation, and stem cell niche were measured. Mice survival was monitored during NEC induction, and ten pups for each group were analysed.

## Intestinal histology

As previously described [16], terminal ileum were placed into 4% paraformaldehyde, dehydrated, and paraffin-embedded for sectioning to 5μm thickness. Once sections were placed on slides, they were stained with hematoxylin and eosin and fixed using SHURMOUNT (General

Data). Photographs were taken of all samples and were blindly scored by three independent reviewers using a well-documented scoring system: Grade 0 = intact/normal structure of villi; 1 = disarrangement of enterocytes and mild villous core separation; 2 = disarrangement of enterocytes and severe villous core separation; and 3 = epithelial sloughing (17). A score of $\geq 2$ was defined as NEC, and a score of $\geq 3$ was defined as severe.

## Quantitative reverse transcription polymerase chain reaction (RT-qPCR)

RNA was isolated from ileal samples using Trizol (TRIzol[TM] Reagent, Invitrogen[TM]) following Invitrogen's recommended protocol [16]. RNA was purified and quantified via NanoDrop[TM] spectrophotometer (Thermofisher Scientific), and 1μg of RNA was utilized for cDNA synthesis (qScript cDNA Supermix, Quantabio). RT-qPCR experiments used SYBR[TM] Green Master Mix (Wisent) for 40 cycles (denaturation = 95˚C; annealing = 58˚C; extension = 72˚C) using primer sequences found in Table 1. ΔΔCT method was used to normalize the gene expression of all cytokines used in this experiment. As it is well documented that interleukin-6 (IL-6) and tumor necrosis factor-α (TNF-α) are important inflammatory markers in NEC [21], we chose these to confirm inflammation. We further investigate whether hAFSC-CM might have any effects on intestinal regeneration or recovery of the stem cell niche following intestinal injury. We measured the intestinal stem cell markers leucine-rich repeat-containing G-protein coupled receptor 5 (*Lgr5*) and olfactomedin (Olfm4) [22]. All markers were compared to glyceraldehyde 3-phosphate dehydrogenase (*GAPDH*) and ribosomal protein lateral stalk subunit P0 (*RPL0*). We compared experimental groups to control and created relative normalized gene expression (shown as fold-change). All gene sequences are listed in Table 1.

## Intestinal immunofluorescence

To study the effects of hAFSC-CM on the intestinal inflammation, apoptosis and regeneration, immunofluorescence staining was performed to visualize markers of neutrophile infiltration Myeloperoxidase (MPO) [23], intestinal apoptosis cleaved caspase 3 (CC3), epithelial proliferation (KI67), and intestinal stem cell (OLFM4).

Terminal ileum were placed into 4% paraformaldehyde, dehydrated, and paraffin-embedded for sectioning to 5μm thickness. Once sections were placed on slides, they were washed

**Table 1. Primer sequences used for RT-qPCR experiments.**

| Target | Forward Primer | Reverse Primer |
|---|---|---|
| Gapdh [a] | TGAAGCAGGCATCTGAGGG | CGAAGGTGGAAGAGTGGGAG |
| Rplo [b] | GGCGACCTGGAAGTCCAACT | CCATCAGCACCACAGCCTTC |
| Il-6 [c] | CTCTGCAAGAGACTTCCATCCA | AGTCTCCTCTCCGGACTTGT |
| Tnf-α [d] | TCCCCAAAGGGATGAGAAGTT | GCTACAGGCTTGTCACTCGAA |
| Lgr5 [e] | TCAATCCCTGCGCCTAGATG | GGGACGTCTGTGAGAGCATT |
| Olfm4 [f] | AGTGACCTTGTGCCTGCC | CACGCCACCATGACTACA |
| Vegf 164 [g] | GCAGGCTGCTGTAACGATGA | GCATGATCTGCATGGTGATGTT |

[a] Glyceraldehyde 3-phosphate dehydrogenase.

[b] Ribosomal protein LO.

[c] Interleukin-6.

d Tumor necrosis factor-alpha.

e leucine-rish repeat-containing G-protein coupled receptor 5.

f Olfactomedin 4.

g Vascular endothelial growth factor.

**Table 2. Antibodies used for immunofluorescence of the ileum.**

| Antibody | Cell population | Species | Dilution | Company | Catalogue number |
|---|---|---|---|---|---|
| **Primary** KI67/MKI67 | Proliferating cells | Rabbit | 1:500 | Abcam (Cambridge, UK) | Ab15580 |
| **Primary** KI67 | Proliferating cells | Rat | 1:500 | Thermo Fisher Scientific (Waltham, MA, USA) | 14-5698-82 |
| **Primary** OLFM4 | Intestinal stem cells | Rabbit | 1:500 | Cell Signalling Technology (Danvers, MS, USA) | 39141 |
| **Primary** CC3 | Apoptosis | Rabbit | 1:200 | Cell Signalling Technology (Danvers, MS, USA) | 9661 |
| **Primary** MPO | Neutrophils | Mouse | 1:200 | R&D systems (Minneapolis, MN, USA) | MAB3174 |
| **Secondary** Anti-rabbit *IgG* [a] | Green fluorescence (488 nanometer) | Donkey | 1:500 | Cell Signalling Technology (Danvers, MS, USA) | 1834802 |
| **Secondary** Anti-mouse *IgG* [a] | Green fluorescence (488 nanometer) | Goat | 1:1000 | Cell Signalling Technology (Danvers, MS, USA) | 4408 |
| **Secondary** Anti-rat *IgG* [a] | Green fluorescence (488 nanometer) | Goat | 1:1000 | Cell Signalling Technology (Danvers, MS, USA) | 4416 |
| **Secondary** Anti-rabbit *IgG* [a] | Red fluorescence (568 nanometer) | Goat | 1:1000 | Thermo Fisher Scientific (Waltham, MA, USA) | A11036 |

[a]Immunoglobulin G.

with triphosphate buffered saline + tween 20 (TBST). Slides were placed into a pressure cooker filled with fresh 10mM sodium citrate pH6 (Bio Basic Canada Inc.; 2.94g in 1L water) (titrated with 1M Hydrochloric acid). Slides were then taken out, cooled to room temperature, and washed again with TBST. Non-specific blocking was achieved through 5% bovine serum albumin (BSA) + 0.1% Triton for 1 hour at room temperature. After blocking, slides were incubated within a humidifying chamber with the primary antibody overnight at 4˚C. TBST was used to wash slides, and then the secondary antibody was used to visualize positive cells (Table 2). Slides were then mounted with Gold anti-fade reagent. To confirm staining to facilitate accuracy of methods, *DAPI* fluorescence was utilized. All photographs were taken with a Nikon TE-2000 epifluorescence/histology digital microscope (Nikon Instruments Inc., NY, USA). Photographs were taken of all samples and were blindly scored by three independent reviewers and each field was counted for positively fluorescing cells compared to the total number of crypts.

## Statistical analyses

GraphPad Prism 7 (one-way ANOVA, Tukey post-hoc test) was used to compare histology, gene expression, and counting immunofluorescence cells between the three experimental groups. The p-value was set to a significance level of $< 0.05$. All graphics were presented as mean ± standard error of the mean (SEM).

## Results

### CM administration improved NEC mice survival and reduced intestinal injury

To study the beneficial effect of AFSC-CM, we first examined the survival rate of the mice pups exposed to NEC induction. Compared to control group, there was significantly reduced survival in the NEC group at postnatal day 9 (**Fig 2A**). Although CM administration seemed to improve the NEC pup's survival, there was no significant difference in survival when comparing to the control or NEC+PBS group. To determine if AFSC-CM improved NEC-induced small intestinal injury, we assessed the morphology of the intestines using a histological scoring system. NEC-like injury, defined by a histological score of $\geq 2$, was present in 89% of NEC +PBS group compared to 20% of NEC+CM group (**Fig 2B and 2C**). Severe NEC (histological score$\geq 3$) was not present in the NEC+CM group (**Fig 2C**). This suggests that NEC-induced intestinal morphological damage was recovered by CM treatment.

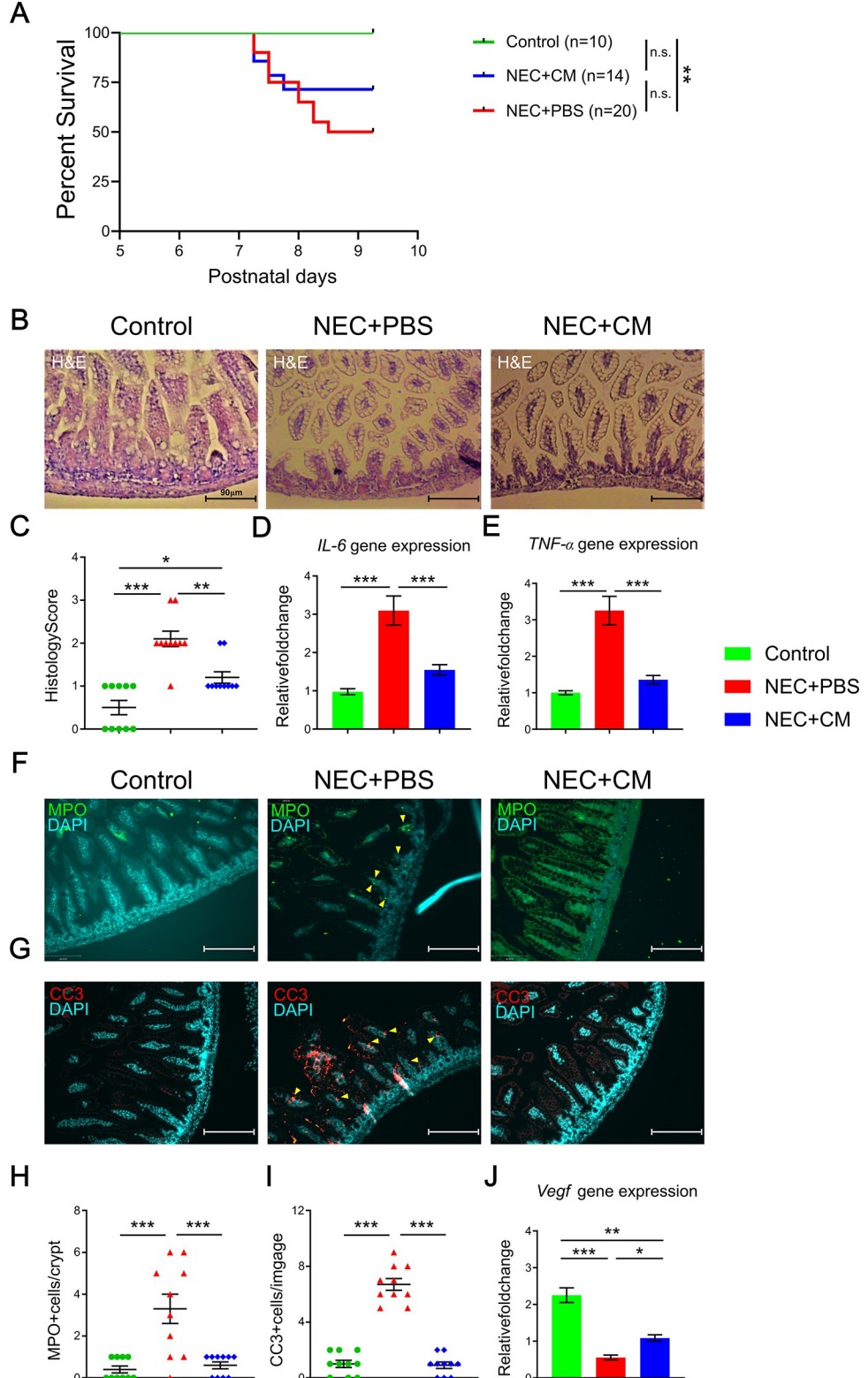

**Fig 2. NEC survival and intestinal injury.** (A) Survival curves for mice pups from the 3 groups on postnatal days 5 to 9. (B) Representative histological images of the terminal ileum. (C) Histological score of intestinal injury. Inflammatory gene marker expression of (D) *Il-6* and (E) *Tnf-α*. Immunofluorescent protein expression of (F) MPO and (G) CC3 in the terminal ilea of the 3 groups. Yellow arrows indicate positive staining. Relative quantification of the

(H) MPO and (I) CC3 protein expressions. (J) Angiogenesis/vascularization *Vegf* gene expression in the 3 groups. Data are presented as mean ± standard error, with significance of group comparisons based on one-way ANOVA and Tukey post-hoc tests. n = 10 for each group, $^*$p<0.05, $^{**}$p<0.01, and $^{***}$p<0.001.

NEC-induced intestinal damage is characterized by increased inflammation and neutrophil infiltration which lead to epithelial cell death. To determine the effect of AFSC-CM on intestinal inflammation we measured the gene expression of intestinal mucosal inflammation markers *Il-6* and *Tnf-α*. The expression of both genes was elevated in NEC group; however, CM administration significantly reduced their expression (**Fig 2D and 2E**). To determine if reduced inflammation was associated with changes in neutrophil activity, we quantified neutrophil infiltration by MPO staining. There was significantly increased MPO positive cell per villi/crypts in the NEC +PBS group relative to control (**Fig 2F and 2H**). CM treatment significantly reduced neutrophil infiltration to the control levels. To illustrate the possible effect of CM on the epithelial apoptosis, we quantified CC3 staining. CC3 positive cells per image showed a significantly increased in the NEC+PBS group relative to control and this was significantly reduced to control levels in the NEC+CM group (p<0.001) (**Fig 2G and 2I**). This indicates that the beneficial effects on AFSC-CM are, at least partially, mediated by reduced inflammatory response during NEC. *Vegf* gene expression was used to evaluate angiogenesis. Compared to control, there was a decrease in *Vegf* expression in NEC+PBS group, and this was restored in the NEC+CM group (**Fig 2J**). These data demonstrate that CM administration improved intestinal injury, mucosal inflammation, epithelial apoptosis, and intestinal angiogenesis.

## Intestinal proliferation and regeneration were increased with CM administration

We have previously demonstrated that NEC induced impairment of intestinal regeneration ability could be restored by the exogenous AFSC cells administration [19]. We hypothesized these effects could be carried out by the AFSC-CM. The intestinal regeneration was assessed by immunofluorescence co-staining of epithelial proliferation (KI67) and intestinal stem cells (OLFM4) (**Fig 3A–3D**). Both epithelial proliferation and intestinal stem cells were decreased at the bottom of crypts in NEC+PBS mice compared to controls but were rescued by CM administration (**Fig 3A–3E**). Similarly, intestinal stem cell markers (*Olfm4*; *Lrg5*) gene expression were significantly decreased in NEC mice, but were rescued by CM treatment (**Fig 3F and 3G**). These data demonstrate that intestinal proliferation and regeneration were impaired in the NEC mice, but re-established by CM administration.

## The secretome of human AFSC-derived CM

To gain a better understanding on the proteins in CM which may mediate the beneficial effects observed in our NEC model, we performed proteomic analysis on CM. We identified 1,500 proteins in CM. As expected, these proteins were most highly enriched by extracellular exosomes by cellular component gene ontology (**Fig 4A**). Functional analysis on these proteins identified several clusters including immune-regulation, cell cycle and stem cell regulation (**Fig 4B**). These functional categories are consistent with the immune and stem cell regulatory properties of CM as identified in our study.

## Discussion

Administration of conditioned medium derived from human amniotic fluid stem cells (hAFSC-CM) reduced the intestinal damage associated with experimental NEC. We

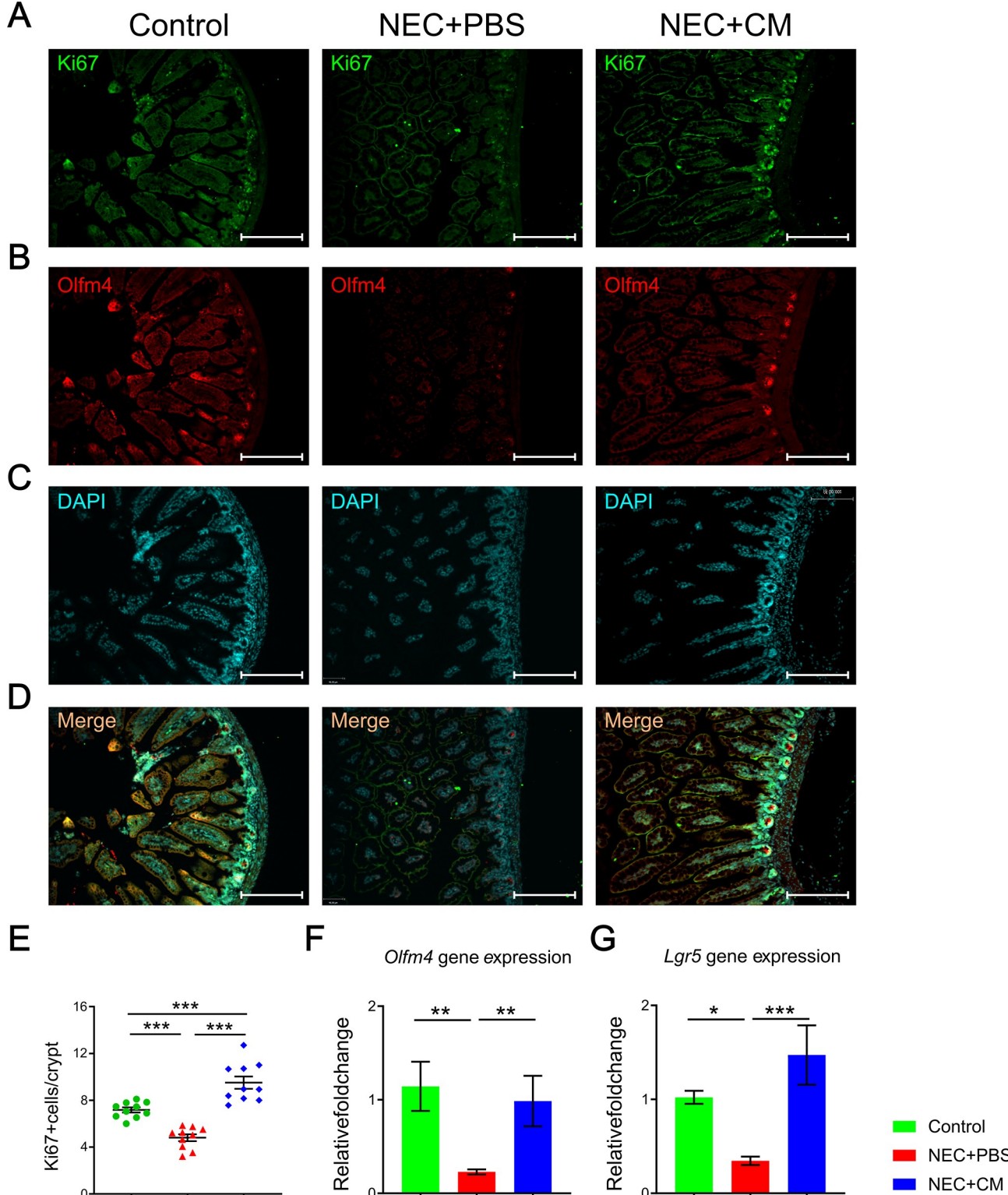

**Fig 3. Intestinal proliferation and regeneration.** Representative immunofluorescent histomicrographs of the terminal ileum from the control, NEC +PBS, and NEC+CM groups, showing expression of **(A)** KI67, **(B)** OLFM4, **(C)** DAPI staining denoting the nuclei for each respective group, and **(D)** merging the 3 stains with yellow staining denoting positive KI67 and DAPI overlap. Relative quantitative expression for **(E)** KI67, **(F)** *Olfm4*, and **(G)** *Lgr5* in the 3 groups. Data are presented as mean ± standard error, with significance of group comparisons based on one-way ANOVA and Tukey post-hoc tests. n = 10 for each group, *p<0.05, **p<0.01, and ***p<0.001.

# A

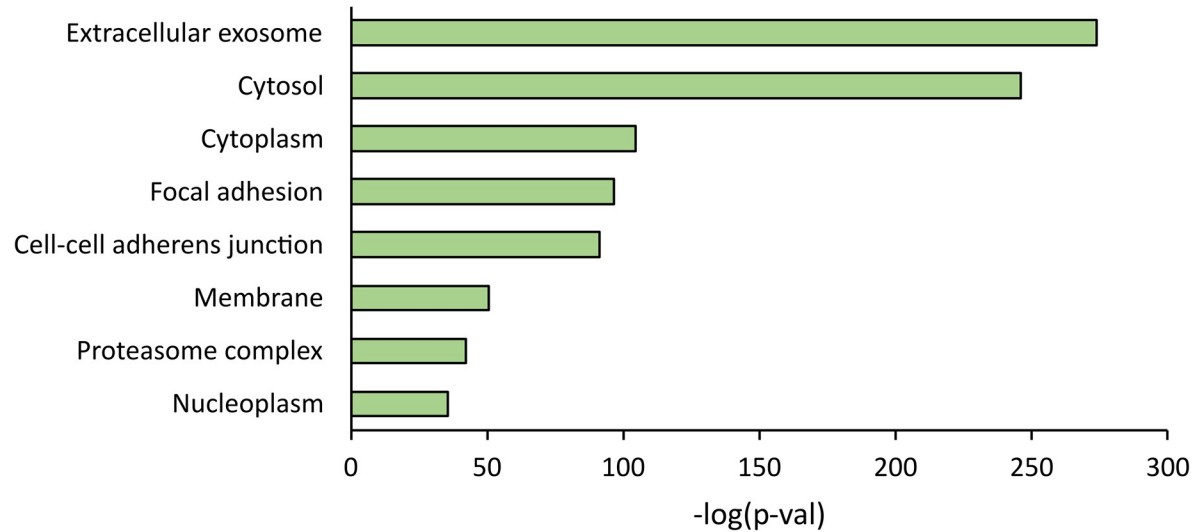

# B

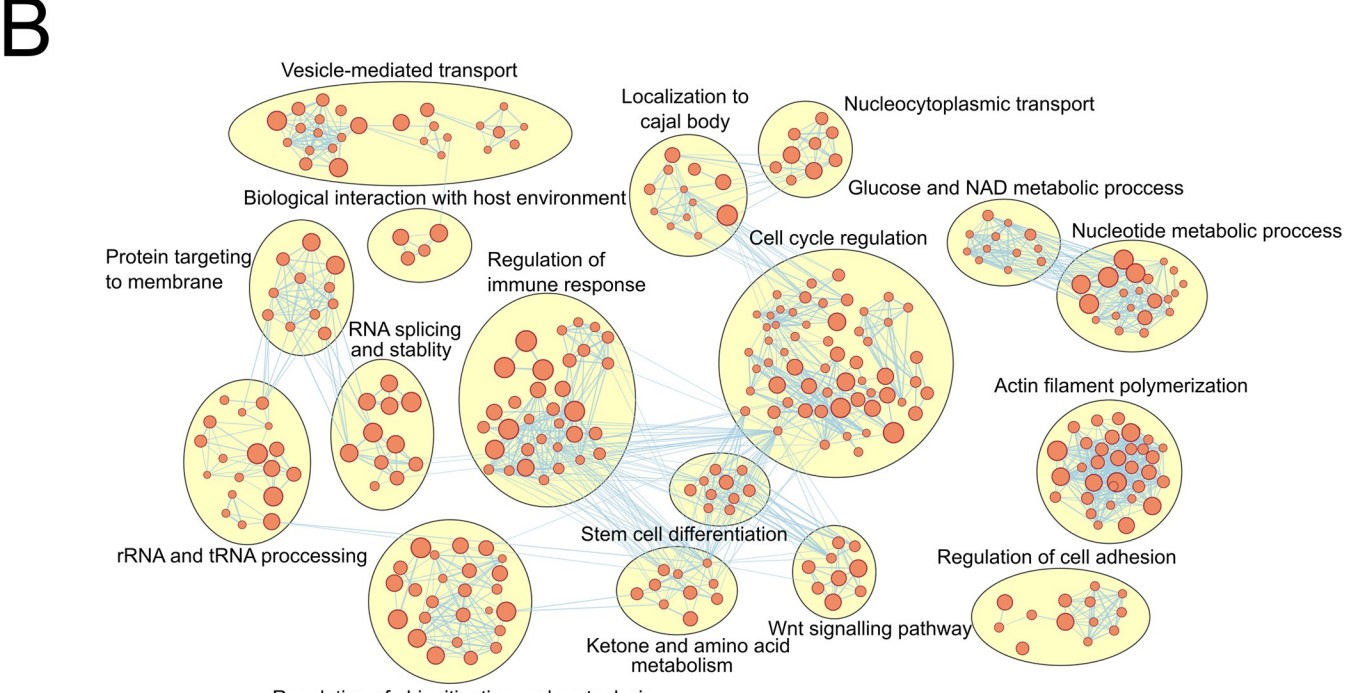

**Fig 4. Proteomic analysis of *human AFSC-derived CM*. (A)** cellular component gene ontology enriched in the secretome of hAFSC-CM. **(B)** Network clustering showing the clusters of biological processes (from gene ontology) enriched in the secretome of hAFSC-CM.

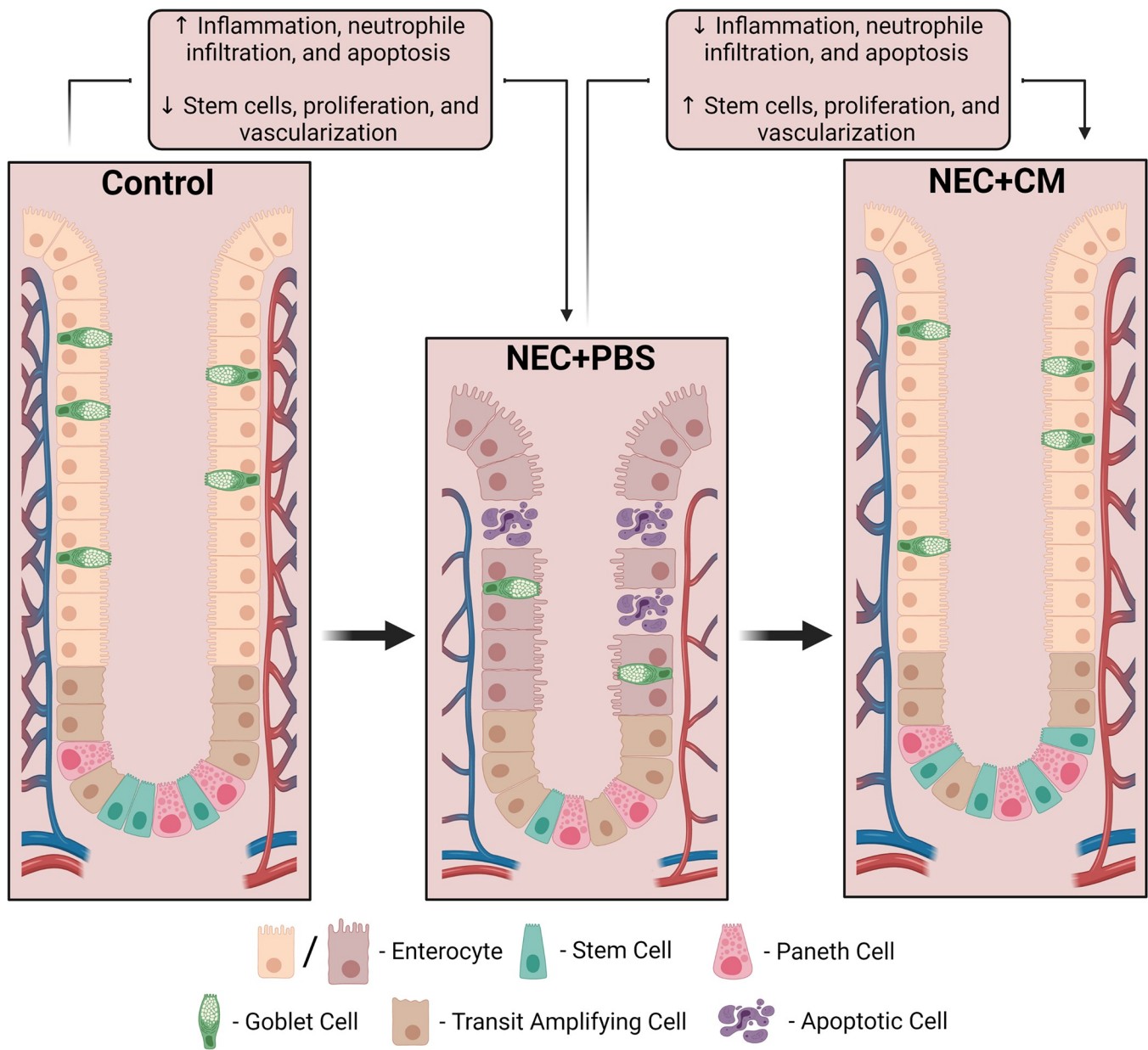

**Fig 5. Summary of the role of AFSC-CM in experimental NEC.** NEC induced an intestinal damage, increased inflammation (IL-6 and TNF-α), neutrophil infiltration (MPO), and elevated apoptosis (CC3), as well as decreased angiogenesis or vascularization (VEGF). Further, it diminished epithelial proliferation (Ki67) and stem cell regeneration activity (Lgr5 and Olfm4), however, CM administration reverses all of these effects to yield reduced intestinal injury and improved survival.

demonstrated that during NEC, hAFSC-CM decrease intestinal damage, reduce ileal inflammation, increase proliferation of intestinal epithelial cells, and recover the stem cell niche within the crypts of the villi and restore the intestinal angiogenesis/microvasculature. This indicates that in experimental NEC, the administration of hAFSC-CM provided numerous benefits (**Fig 5**).

Stem cells, like those derived from amniotic fluid or adult bone marrow, have been studied and utilized in this disease model and has showed anti-inflammatory and pro-regenerative potentials. In NEC, investigations have been trying to elucidate which type of stem cell is

superior and how they can best be used for improving intestinal recovery [4, 6, 7, 9, 24]. While stem cell therapy is desirable due to benefits stated previously, a cellular-free product with similar potential would be more desirable [4, 9, 24]. Although the mechanism of stem cells is not completely understood, a proposed mechanism is the ability to release acellular factors in a paracrine fashion to allow healing, recovery, and repopulation [4]. The two most common cell lineages used in relation to NEC are of fetal (AFSC derived from amniotic fluid) and postnatal (BM-MSC from adult bone marrow) origin. While the latter could potentially be used for treatment [6–8]. AFSC seem to be overall superior based on ease of cellular proliferation and reduced tumorgenicity [6, 10]. AFSC are also the preferable type of stem cell due to being safe for collection (amniocentesis). Moreover, AFSC can multiply more rapidly compared to BM-MSC [6, 10]. which is important for any fast and effective treatment.

The administration of AFSC in infants with NEC present several logistic difficulties in relation to the dose, route of administration and potential short- and long-term complications. These would include the possibility of tumor growth later in life, the potential immune reactions related to the stem cells grafting to the patients, and unknown long-term effects [9]. Stem cells derived from amniotic fluid are pluripotent and have not shown any tumorigenic potential [11]. With these inherent attributes, there would be an advantage of using hAFSC-CM as they retain the vital treatment properties of AFSC (as seen in this study) while lacking cellular products or other immune complexes that may cause negative reactions when transplanting or injecting. This suggests it may be possible to use hAFSC-CM for therapy with minimal drawbacks compared to injecting stem cells directly [13].

For this reason, we focused our attention on CM derived from AFSC as the benefits of AFSC administration seem to be related to a paracrine action instead of repopulation of the injured intestinal epithelium [4]. Our study showed that hAFSC-CM administration is associated with increased stem cell activity and recovery from NEC. This is similar to the benefits obtained by AFSC administration in a similar NEC model [4]. To deepen our study on the recovery of intestinal tissue during NEC, we measured and visualized neutrophile infiltration (MPO), cell apoptosis (CC3), proliferation (Ki67), and native stem cell activity (Olfm4; Lgr5) in the ileal crypts. Following hAFSC-CM treatment, we noticed reduced apoptosis and neutrophile infiltration in the hAFSC-CM treated group compared to NEC+PBS. Furthermore, there was recovery of proliferation and stem cell activity in hAFSC-CM treated pups compared to NEC+PBS. It has been reported that the reduction of VEGF led to a failure in development and maintenance of capillary networks, which contributed to the NEC pathogenesis [25]. We have demonstrated treatment with CM restores VEGF levels in the current study.

Although hAFSC-CM holds vital cellular information found in AFSC such as mRNA, microRNA, DNA, proteins, and extracellular vesicles [13], the mechanism of action of CM in our disease model has yet to be fully understood. It has been suggested that AFSC administration works by activating paracrine signalling between cells [4, 9]. It is possible that CM utilizes this same pathway as it is known to hold important cell signalling, proteins, and nuclear information similar to AFSC [13]. hAFSC-CM shows promise in the prevention and treatment of experimental NEC while lacking cellular products known to produce detrimental side effects from AFSC. The constituents of the CM produced using the protocol of this study differs to the secretome produced by AFSC cells grown under standard conditions (containing FCS). Nevertheless, the beneficial effect from AFSC-CM could be endorsed by the functional breakdown of the secreted protein functions (**Fig 4**), since major clusters are noted such as immune-regulation, cell cycle and stem cell regulation. Interestingly, another large functional cluster of the secretome is vesicle mediated transport, indicating a further study to identify the cargo in the extracellular vesicles (protein and miRNA) is necessary [13]. As we previously demonstrated the AFSC derived EVs could be beneficial in reducing the NEC induced

intestinal damage [16], which is similar to the AFSC CM effects we found in this study. However, AFSC-CM components do not only contain EVs, but also has proteins that could represent broader effects than the AFSC EV. That is the rational for our current study to lay down the foundation to mining the beneficial component from AFSC-CM.

In addition to the signalling pathway remaining unknown, we are also unaware of which specific components of CM have a pronounced effect on the recovery from NEC related damage. Investigators feel that nanoparticles (i.e. microvesicles, secretomes, and exosomes) within the CM may hold the key information needed to recover damaged tissue, although this hypothesis has yet to be substantiated [6, 13, 26]. It is important to determine which components within the CM are responsible for reducing the effects of NEC so that appropriate dosages and concentrations may be developed. In this experiment, we injected 100μL of CM into a 3.5g mouse pup twice over this NEC model. This would be the equivalent of injecting >100mL of CM into a neonate. Instead of administering this large amount of fluid, it would be best practice to identify which components within the CM are most useful and then administer it in a concentrated form. This would likely reduce complications from overdosing and avoid administering unnecessary components. Although it is unlikely that any part of the CM is immunogenic or will cause adverse effects (like those seen in AFSC or BM-MSC), it is best practice to use the lowest dose for the best overall result.

During the last 20 years, no medical advancement has been made in the treatment of NEC to avoid its progression to intestinal necrosis or perforation [9]. hAFSC-CM administration is an exciting novel treatment for infants with NEC. Our experiment included intra-peritoneal injection to simulate intravenous injection and ascertain deliver in the intestinal circulation. This is based on the hypothesis that AFSC (and CM derived from these cells) would carry important cellular markers and proteins to salvage damaged bowel through paracrine signalling. However, further studies are needed to characterise the ideal dose of CM, the optimal route of administration, as well as determine which specific components of CM are responsible for salvaging the intestine from NEC damage. Beyond these measures, it will be important to investigate any short- or long-term complications associated with this innovative treatment.

## Conclusions

When administered during experimental NEC, hAFSC-CM reduced injury to the terminal ileum. We observed that, compared to phosphate buffered saline (placebo), there was reduced histological injury, reduced inflammatory cytokines, increased intestinal stem cell expression, and an increased intestinal enterocyte proliferation. Further experimentation is required to elucidate how this may be used as a treatment. This experiment provides the scientific foundation needed for potential translation into humans.

## Supporting information

**S1 Data.**
(XLSX)

## Author Contributions

**Conceptualization:** Joshua S. O'Connell, Simon Eaton, Robert Mitchell, Paolo De Coppi, Ketan Patel, Agostino Pierro.

**Data curation:** Joshua S. O'Connell, Andrea Zito, Abdalla Ahmed, Marissa Cadete, Niloofar Ganji, Ethan Lau, Mashriq Alganabi, Nassim Farhat, Carol Lee, Steve Ray.

**Formal analysis:** Abdalla Ahmed.

**Funding acquisition:** Agostino Pierro.

**Investigation:** Bo Li.

**Supervision:** Agostino Pierro.

**Visualization:** Andrea Zito.

**Writing – original draft:** Andrea Zito.

**Writing – review & editing:** Mashriq Alganabi.

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
