## [Decision Letter · Decision Letter 0]

27 Apr 2021

PONE-D-21-06129

Treatment of necrotizing enterocolitis by conditioned medium derived from human amniotic fluid stem cells

PLOS ONE

Dear Dr. Pierro

Thank you for submitting your manuscript to PLOS ONE. After careful consideration, we feel that it has merit but does not fully meet PLOS ONE’s publication criteria as it currently stands. Therefore, we invite you to submit a revised version of the manuscript that addresses the points raised during the review process.

In particular, the results of this study are of interest . However, the manuscript would increase merit if the potential mechanism of action of the CM in respect to cell therapy would be better detailed. Moreover, the identification of the CM dosage chosen needs rationale.

We look forward to receiving your revised manuscript.

Kind regards,

Benedetta Bussolati, MD, PhD

Academic Editor

PLOS ONE

Journal Requirements:

[AP is the recipient of a Canadian Institutes of Health Research (CIHR) Foundation Grant 353857. The funders had no role in study design, data collection and analysis, decision to publish, or preparation of the manuscript.].    

We note that one or more of the authors are employed by a commercial company: Micregen Ltd.

Reviewers' comments:

Reviewer's Responses to Questions

**Comments to the Author**

1. Is the manuscript technically sound, and do the data support the conclusions?

Reviewer #1: Partly

Reviewer #2: Yes

2. Has the statistical analysis been performed appropriately and rigorously? 

Reviewer #1: Yes

Reviewer #2: Yes

3. Have the authors made all data underlying the findings in their manuscript fully available?

Reviewer #1: Yes

Reviewer #2: Yes

4. Is the manuscript presented in an intelligible fashion and written in standard English?

Reviewer #1: Yes

Reviewer #2: Yes

5. Review Comments to the Author

Reviewer #1: Joshua S. O’Connell et al. present a research on therapeutic effect of conditioned medium (CM) derived from human AFSC on NEC without risk of tumorgenesis as well as immunogenicity. The manuscript must describe a technically sound piece of scientific research with data that partly supports the conclusions. Experiments must have been conducted rigorously, with appropriate controls, replication, and sample sizes. The conclusions must be drawn appropriately based on the data presented.It is quite interesting but remains some problems that needs to be addressed.

Detailed comments are as follows:

1. In line 61, what kinds of MSC does the author refer to? As postnatal MSC contain various types of MSC such as bone marrow-derived or adipose-derived mesenchymal stem cells.

2. Flow chart of the experiment is suggested.

3. Why does the author choose 100μL as the dose of CM to treat NEC? The author should explain.

4. Is there any difference of therapeutic efficacy between CM and MSC transplantation?

5. The author should add more experiments to examine the effect of CM on NEC such as apotosis or others.

6. As the effect of CM was so prominent, the author need to explain what composition in CM plays an key role in the CM-mediated therapeutic effect.

Reviewer #2: This manuscript by O’Connell JS et al. describes the protective effects of conditioned medium derived from human amniotic fluid stem cells (AFSC) in experimental mouse model of necrotizing enterocolitis (NEC). NEC was induced starting at post-natal day P5 via gavage feeding of hyperosmolar formular 4x a day, immediately following 10 minute hypoxia prior to all feeds and LPS administration on P6 and P7. For treatment, CM were administered on P6 and P7 at a dose of 100uL/each via IP injection; PBS administration were used as control. The experiment was terminated 2 days after the last administration of CM at post-natal day P9, and terminal ileum were harvested and processed for histologic evaluation and molecular characterization for IL-6, TNFalpha, Lgr5 gene expression. The authors concluded that the CM treatment group demonstrated less tissue damage, lower pro-inflammatory IL-6 and TNF-alpha expression and improved counts for stem cell activity marker Lgr5. The manuscript is well written and the research study address an important problem in neonatal gastroenterology. Knowledge gained from this study could potentially advance the field.

Major concerns:

1. The authors have previously published (ref. #14) that human AFSC derived extracellular vesicles provide benefits identical to the results reported herein Are the AFSC used in this study different from that used in reference #14, and if so, how? The authors refer to ref. #14 in relation to methods used and in the context of EVs in general as a potential mechanism of action in the CM, but the discussion falls short of clearly stating that those EVs were also derived from AFSC. It should be therefore clearly stated in the discussion how these two studies are different in terms of the stem cells used (if they are), otherwise it seems as if the hypothesis proposed herein has been previously tested. If the stem cells in both studies are of the same kind (not necessarily from the same source), then there is no novelty and/or significance to the present study, as in some respects it replicates previous work.

2. The data for gene expression for IL6, TNFa and Lgr5 are impressive, however, confirmation of this effect at protein level will be more convincing. VEGF is known to be decreased during NEC; have you assessed if treatment with CM helps restore VEGF levels? How about neutrophils and macrophages?

3. Also, adding an immunostaining for Lgr5 or rather a co-staining of Lgr5 and Ki67 together will be more valuable and informative to demonstrate intestinal repair and regeneration.

4. How was it determined that two doses of CM administration were needed to generate the beneficial effect? Is there evidence that single dose is not effective?

5. Does the CM treatment provide long term benefit, or the observed changes are transient? In this regard, a survival study will be very useful for demonstrating any lasting effects.

Minor concerns:

1. DAPI staining (described in methods) is missing in Figure 4.

6. PLOS authors have the option to publish the peer review history of their article (what does this mean?). If published, this will include your full peer review and any attached files.

Reviewer #1: No

Reviewer #2: No

---

## [Author Response · Author response to Decision Letter 0]

27 Sep 2021

Response to Reviewers’ Comments

Treatment of necrotizing enterocolitis by conditioned medium derived from human amniotic fluid stem cells

We would like to thank the reviewers for their careful review of our manuscript, and their thoughtful and constructive comments. Our responses to the reviewers’ comments are in blue in this letter. The corresponding modifications in the manuscript are highlighted in yellow.

Reviewer #1:

Joshua S. O’Connell et al. present a research on therapeutic effect of conditioned medium (CM) derived from human AFSC on NEC without risk of tumorgenesis as well as immunogenicity. The manuscript must describe a technically sound piece of scientific research with data that partly supports the conclusions. Experiments must have been conducted rigorously, with appropriate controls, replication, and sample sizes. The conclusions must be drawn appropriately based on the data presented. It is quite interesting but remains some problems that needs to be addressed.

Detailed comments are as follows:

1. In line 61, what kinds of MSC does the author refer to? As postnatal MSC contain various types of MSC such as bone marrow-derived or adipose-derived mesenchymal stem cells.

Response: 

We agree with the reviewer that this needed to be clarified. The postnatal MSCs we refer to are bone marrow derived. This change has been made in the introduction of the manuscript (Line 61).

2. Flow chart of the experiment is suggested.

Response: 

Thank you for this suggestion. A flow chart has been added to the discussion of the manuscript. 

3. Why does the author choose 100μL as the dose of CM to treat NEC? The author should explain.

Response: 

The chosen dose of 100μL is based on previous publications of conditioned media from human amniotic fluid-derived stem cells (hAFSC-CM) and their therapeutic effects in experimental mice models1,2. This information has been added to the materials and methods of the manuscript (Lines 137-139). 

4. Is there any difference of therapeutic efficacy between CM and MSC transplantation?

Response: 

There is no difference of therapeutic efficacy between CM and MSC transplantation. Similar to our results, it has been reported in experimental NEC that IP injection of bone marrow-derived mesenchymal stem cells (BM-MSC) and amniotic fluid-derived mesenchymal stem cells (AF-MSC) significantly reduced NEC incidence and morphological damage3,4. As demonstrated in this experimental NEC study, AFSC-CM administration reduced NEC incidence and morphological damage, inflammation (myeloperoxidase), and cell apoptosis (cleaved caspase 3), as well as increased intestinal proliferation and stem cell activation. These results are similar to what was found with IP injection of AFSCs in experimental NEC5.

5. The author should add more experiments to examine the effect of CM on NEC such as apotosis or others.

Response: 

We have performed additional experiments to study the beneficial effects of CM on the intestinal inflammation, apoptosis, and angiogenesis. This information has been added to the manuscript (Lines 221-235).

6. As the effect of CM was so prominent, the author need to explain what composition in CM plays an key role in the CM-mediated therapeutic effect.

Response: 

We have performed additional experiments to study the proteomics of the AFSC-CM. This information has been added to the manuscript (Lines 269-275).

Reviewer #2:

This manuscript by O’Connell JS et al. describes the protective effects of conditioned medium derived from human amniotic fluid stem cells (AFSC) in experimental mouse model of necrotizing enterocolitis (NEC). NEC was induced starting at post-natal day P5 via gavage feeding of hyperosmolar formular 4x a day, immediately following 10 minute hypoxia prior to all feeds and LPS administration on P6 and P7. For treatment, CM were administered on P6 and P7 at a dose of 100uL/each via IP injection; PBS administration were used as control. The experiment was terminated 2 days after the last administration of CM at post-natal day P9, and terminal ileum were harvested and processed for histologic evaluation and molecular characterization for IL-6, TNFalpha, Lgr5 gene expression. The authors concluded that the CM treatment group demonstrated less tissue damage, lower pro-inflammatory IL-6 and TNF-alpha expression and improved counts for stem cell activity marker Lgr5. The manuscript is well written and the research study address an important problem in neonatal gastroenterology. Knowledge gained from this study could potentially advance the field. 

Major concerns:

1. The authors have previously published (ref. #14) that human AFSC derived extracellular vesicles provide benefits identical to the results reported herein Are the AFSC used in this study different from that used in reference #14, and if so, how? The authors refer to ref. #14 in relation to methods used and in the context of EVs in general as a potential mechanism of action in the CM, but the discussion falls short of clearly stating that those EVs were also derived from AFSC. It should be therefore clearly stated in the discussion how these two studies are different in terms of the stem cells used (if they are), otherwise it seems as if the hypothesis proposed herein has been previously tested. If the stem cells in both studies are of the same kind (not necessarily from the same source), then there is no novelty and/or significance to the present study, as in some respects it replicates previous work.

Response: 

We fully agree with the reviewer’s comments. As we previously demonstrated the AFSC derived EVs could be beneficial in reducing the NEC induced intestinal damage6, which is similar to the AFSC CM effects we found in this study. However, the AFSC-CM contains components not limited exclusively to EVs, but also proteins, indicating a broader effect than exclusive AFSC EVs administration. That is the rational for our current study to lay down the foundation to mining the beneficial component from AFSC CM. This information has been added to the manuscript (lines 345-349).

2. The data for gene expression for IL6, TNFa and Lgr5 are impressive, however, confirmation of this effect at protein level will be more convincing. VEGF is known to be decreased during NEC; have you assessed if treatment with CM helps restore VEGF levels? How about neutrophils and macrophages?

Response: 

We have performed additional experiments to study the beneficial effects of CM on intestinal inflammation, neutrophil infiltration, apoptosis, and angiogenesis. This information has been added to the manuscript (Lines 221-235).

3. Also, adding an immunostaining for Lgr5 or rather a co-staining of Lgr5 and Ki67 together will be more valuable and informative to demonstrate intestinal repair and regeneration.

Response: 

Immunostaining for Lgr5 is difficult due to no validated specific antibody. Instead, we performed co-staining for intestinal stem cell marker Olfm47, and epithelial proliferation marker Ki67 to demonstrate the intestinal regeneration ability. The new experimental data has been added this manuscript (Lines 248-257)

4. How was it determined that two doses of CM administration were needed to generate the beneficial effect? Is there evidence that single dose is not effective?

Response: 

We agree with the reviewer that this needs to be clarified. The two-dose administration was based on previous publications that showed therapeutic effects of IP injected AFSC5 and AFSC extracellular vesicles8 in experimental NEC. Two doses via similar administration of IP injection allows for comparison with respect to potential treatment.

This information has been added to the materials and methods of the manuscript (Lines 137-139). 

5. Does the CM treatment provide long term benefit, or the observed changes are transient? In this regard, a survival study will be very useful for demonstrating any lasting effects.

Response: 

We have performed additional experiments to study the mice pup’s survival. This new data has been added to the manuscript (Lines 211-215).

Minor concerns:

1. DAPI staining (described in methods) is missing in Figure 4.

Response: 

This information has been added to Figure 4 of the manuscript.

References for the response for the reviewers:

1. Jun EK, Zhang Q, Yoon BS, et al. Hypoxic conditioned medium from human amniotic fluid-derived mesenchymal stem cells accelerates skin wound healing through TGF-β/SMAD2 and PI3K/Akt pathways. Int J Mol Sci. 2014;15(1):605-628. Published 2014 Jan 6. 

2. Mellows B, Mitchell R, Antonioli M, et al. Protein and Molecular Characterization of a Clinically Compliant Amniotic Fluid Stem Cell-Derived Extracellular Vesicle Fraction Capable of Accelerating Muscle Regeneration Through Enhancement of Angiogenesis. Stem Cells Dev. 2017;26(18):1316-1333. 

3. McCulloh, C. J., Olson, J. K., Zhou, Y., Wang, Y., & Besner, G. E. Stem cells and necrotizing enterocolitis: A direct comparison of the efficacy of multiple types of stem cells. Journal of Pediatric Surgery. 2017;52(6):999–1005. 

4. Tayman, C., Uckan, D., Kilic, E., Ulus, A. T., Tonbul, A., Murat Hirfanoglu, I., Helvacioglu, F., Haltas, H., Koseoglu, B., & Tatli, M. M. Mesenchymal Stem Cell Therapy in Necrotizing Enterocolitis: A Rat Study. Pediatric Research. 2011;70(5):489–494. 

5. Zani A, Cananzi M, Fascetti-Leon F, et al. Amniotic fluid stem cells improve survival and enhance repair of damaged intestine in necrotising enterocolitis via a COX-2 dependent mechanism. Gut. 2014;63(2):300-309. 

6. O’Connell JS, Lee C, Farhat N, Antounians L, Zani A, Li B, Pierro A. Administration of extracellular vesicles derived from human amniotic fluid stem cells: a new treatment for necrotizing enterocolitis. Pediatr Surg Int. 2021. 

7. van der Flier LG, Haegebarth A, Stange DE, van de Wetering M, Clevers H. OLFM4 is a robust marker for stem cells in human intestine and marks a subset of colorectal cancer cells. Gastroenterology. 2009;137(1):15-17. 

8. Li B, Lee C, O'Connell JS, Antounians L, Ganji N, Alganabi M, Cadete M, Nascimben F, Koike Y, Hock A, Botts SR, Wu RY, Miyake H, Minich A, Maalouf MF, Zani-Ruttenstock E, Chen Y, Johnson-Henry KC, De Coppi P, Eaton S, Maattanen P, Delgado Olguin P, Zani A, Sherman PM, Pierro A. Activation of Wnt signaling by amniotic fluid stem cell-derived extracellular vesicles attenuates intestinal injury in experimental necrotizing enterocolitis. Cell Death Dis. 2020 Sep 14;11(9):750.

---

## [Decision Letter · Decision Letter 1]

13 Oct 2021

PONE-D-21-06129R1Treatment of necrotizing enterocolitis by conditioned medium derived from human amniotic fluid stem cellsPLOS ONE

Dear Dr.Pierro

Thank you for submitting your manuscript to PLOS ONE. After careful consideration, we feel that it has merit but does not fully meet PLOS ONE’s publication criteria as it currently stands. Therefore, we invite you to submit a revised version of the manuscript that addresses the points raised during the review process. In particular, although the manuscript is ameliorated, it still requires an improvement in the quality of data presentation and discussion,  typos to be corrected and figures to be referred in the text.

We look forward to receiving your revised manuscript.

Kind regards,

Benedetta Bussolati, MD, PhD

Academic Editor

PLOS ONE

Journal Requirements:

Reviewers' comments:

Reviewer's Responses to Questions

**Comments to the Author**

1. If the authors have adequately addressed your comments raised in a previous round of review and you feel that this manuscript is now acceptable for publication, you may indicate that here to bypass the “Comments to the Author” section, enter your conflict of interest statement in the “Confidential to Editor” section, and submit your "Accept" recommendation.

Reviewer #1: All comments have been addressed

Reviewer #2: All comments have been addressed

2. Is the manuscript technically sound, and do the data support the conclusions?

Reviewer #1: Yes

Reviewer #2: Yes

3. Has the statistical analysis been performed appropriately and rigorously? 

Reviewer #1: Yes

Reviewer #2: Yes

4. Have the authors made all data underlying the findings in their manuscript fully available?

Reviewer #1: Yes

Reviewer #2: Yes

5. Is the manuscript presented in an intelligible fashion and written in standard English?

Reviewer #1: Yes

Reviewer #2: No

6. Review Comments to the Author

Reviewer #1: (No Response)

Reviewer #2: This revised manuscript shows an improvement over the original submission, and the new data provided greatly enhances the interpretation of the results. However, the manuscript will further benefit by addressing the following:

1. Figure 2F-G are not referenced in the text.

2. Grammatical error in the following lines 230-232

3. Figure 2J is not referenced. The VEGF expression data is missing in the results. What is the conclusion from the VEGF data is not mentioned throughout the manuscript.

4. The DAPi staining in Figure 3C for the NEC +PBS group does not match with the rest of the stainings.

5. The sample size for the NEC + PBS group needs clarification. It is n=11 according to the dot plot in Figure 3E, but the legend states n=10. This needs to be clarified also in the methods, line 141-142.

6. Figure 5 illustration can be improved by providing the following:

a. description of the cells types shown

b. change of gene or pathway expression in the NEC +CM group similar to the NEC (or NEC + PBS)

7. PLOS authors have the option to publish the peer review history of their article (what does this mean?). If published, this will include your full peer review and any attached files.

Reviewer #1: No

Reviewer #2: **Yes: **Sargis Sedrakyan

---

## [Author Response · Author response to Decision Letter 1]

21 Oct 2021

Response to Reviewers’ Comments

Treatment of necrotizing enterocolitis by conditioned medium derived from human amniotic fluid stem cells

We would like to thank the reviewers for their careful review of our manuscript, and their thoughtful and constructive comments. Our responses to the reviewers’ comments are in blue in this letter. The corresponding modifications in the manuscript are highlighted in yellow.

Reviewer #2: This revised manuscript shows an improvement over the original submission, and the new data provided greatly enhances the interpretation of the results. However, the manuscript will further benefit by addressing the following: 

1. Figure 2F-G are not referenced in the text.

Response: 

Figure 2 H and I are the quantifications of Figure 2 F and G. We have updated our text (Line 228 and 232).

2. Grammatical error in the following lines 230-232

Response: 

Thank you for highlighting this error. We have updated our text, lines 230-232.

3. Figure 2J is not referenced. The VEGF expression data is missing in the results. What is the conclusion from the VEGF data is not mentioned throughout the manuscript.

Response: 

We have included the result of the VEGF expression in the “Result” and “Discussion” sections. See lines 233-235 and lines 289, 330-332

4. The DAPi staining in Figure 3C for the NEC +PBS group does not match with the rest of the stainings.

Response: 

Figure 3C and D has been updated.

5. The sample size for the NEC + PBS group needs clarification. It is n=11 according to the dot plot in Figure 3E, but the legend states n=10. This needs to be clarified also in the methods, line 141-142.

Response: 

Thank you for pointing out this mistake. We have updated the NEC+PBS group in Figure 3E.

6. Figure 5 illustration can be improved by providing the following:

a. description of the cells types shown

b. change of gene or pathway expression in the NEC +CM group similar to the NEC (or NEC + PBS)

Response: 

Figure 5 has been updated to include the description of cell types and the pathways expressed in the NEC+PBS and NEC+CM groups.

---

## [Editor Report · Decision Letter 2]

12 Nov 2021

Treatment of necrotizing enterocolitis by conditioned medium derived from human amniotic fluid stem cells

PONE-D-21-06129R2

Dear Dr. Pierro,

We’re pleased to inform you that your manuscript has been judged scientifically suitable for publication and will be formally accepted for publication once it meets all outstanding technical requirements.

Kind regards,

Benedetta Bussolati, MD, PhD

Academic Editor

PLOS ONE
---

## [Editor Report · Acceptance letter]

25 Nov 2021

PONE-D-21-06129R2 

Treatment of necrotizing enterocolitis by conditioned medium derived from human amniotic fluid stem cells 

Dear Dr. Pierro:

I'm pleased to inform you that your manuscript has been deemed suitable for publication in PLOS ONE. Congratulations! Your manuscript is now with our production department. 

Kind regards, 

on behalf of

Prof. Benedetta Bussolati 

Academic Editor

PLOS ONE